# A Single-Cell Transcriptome Atlas of Epithelial Subpopulations in HPV-Positive and HPV-Negative Head and Neck Cancers

**DOI:** 10.3390/v17040461

**Published:** 2025-03-24

**Authors:** Mary C. Bedard, Cosette M. Rivera-Cruz, Tafadzwa Chihanga, Andrew VonHandorf, Alice L. Tang, Chad Zender, Matthew T. Weirauch, Robert Ferris, Trisha M. Wise-Draper, Mike Adam, Susanne I. Wells

**Affiliations:** 1Division of Oncology, Cincinnati Children’s Hospital Medical Center, Cincinnati, OH 45229, USA; 2Medical Scientist Training Program, University of Cincinnati College of Medicine, Cincinnati, OH 45267, USA; 3Division of Allergy and Immunology, Cincinnati Children’s Hospital Medical Center, Cincinnati, OH 45229, USA; 4Center for Autoimmune Genomics and Etiology, Cincinnati Children’s Hospital Medical Center, Cincinnati, OH 45229, USA; 5Department of Otolaryngology, University of Cincinnati College of Medicine, Cincinnati, OH 45267, USAzendercd@ucmail.uc.edu (C.Z.); 6Divisions of Human Genetics, Biomedical Informatics, and Developmental Biology, Cincinnati Children’s Hospital Medical Center, Cincinnati, OH 45229, USA; 7UNC Lineberger Comprehensive Cancer Center, UNC Health Care System, Chapel Hill, NC 27599, USA; robert_ferris@med.unc.edu; 8Division of Hematology/Oncology, Department of Internal Medicine, University of Cincinnati College of Medicine, Cincinnati, OH 45267, USA; 9Department of Pediatrics, University of Cincinnati College of Medicine, Cincinnati, OH 45267, USA

**Keywords:** human papillomavirus, HPV, head and neck cancer, HNSCC, single-cell transcriptomics, scRNAseq, epithelial cells, cancer therapy

## Abstract

Persistent infection with HPV causes nearly 5% of all cancers worldwide, including cervical and oropharyngeal cancers. Compared to HPV-negative (HPV−) head and neck squamous cell carcinomas (HNSCCs), HPV-positive (HPV+) HNSCCs exhibit a significantly improved treatment response; however, established treatment regimens were largely developed for HPV− disease. Effectively de-escalating therapy and optimizing treatment protocols to minimize toxicity for both HPV+ and HPV− tumors has been variably successful, in part due to the heterogeneity of cellular subpopulations. Single-cell RNA sequencing (scRNAseq) has primarily been used to define immune cell populations rather than the cell type of origin, epithelial cells. To address this, we analyzed published scRNAseq data of HPV+ and HPV− HNSCCs to distinguish epithelial tumor cell populations as a function of HPV status. We identified the transcriptome signatures, ontologies, and candidate biomarkers of newly identified epithelial subpopulations with attention to those that are shared or enriched in HPV+ or HPV− HNSCCs. We hypothesize that distinct epithelial cell populations and reprogramming in HPV− versus HPV+ HNSCC represent important components of the pro-tumor environment. These are described here as a foundation for the identification of new epithelial-cell-specific biomarkers, effectors, and candidate targets for optimizing the treatment of HNSCC.

## 1. Introduction

Although a causal relationship between HPV infection and head and neck squamous cell carcinoma (HNSCC) development was first reported over two decades ago [1] and a preventive vaccine is available, the incidence of HPV+ HNSCCs has continued to increase and is predicted to represent the majority of HNSCCs in the future [2,3]. This trend reflects the continued prevalence of oral HPV infection (i.e., ~7% of the US population, higher amongst men) [4] that is associated with a 50-fold increased risk for developing HPV+ HNSCC [5]. It is well established that HPV+ and HPV− HNSCCs differ in anatomical distribution, molecular drivers, patient demographics, treatment response, and clinical course, and as such, can be considered distinct cancer types that warrant distinct treatment algorithms [6]. For example, HPV-positive and HPV-negative HNSCCs tend to arise in different anatomical locations: HPV-negative HNSCCs primarily occur in the oral cavity, larynx, and hypopharynx, whereas HPV-positive HNSCCs commonly occur in the oropharynx (e.g., tonsils, base of tongue, soft palate). This distinction aligns with epidemiological data showing that approximately 90% of oral cavity HNSCCs are HPV-negative [7], while around 70% of oropharyngeal HNSCCs are HPV-positive [8]. Altogether, these differences support ongoing efforts to tailor HNSCC treatments based on patient and/or tumor characteristics. For example, window-of-opportunity studies in HPV− HNSCCs utilizing PD1 or PDL1-targeting immune therapy have shown improved outcomes in intermediate-risk HNSCC patients [9] and in locally advanced resectable HNSCC (KEYNOTE-689). However, additional studies are needed to understand why some patients will respond favorably to de-escalation and how to optimize treatment paradigms based on tumor characteristics, for example, by HPV status.

With regards to HPV+ HNSCCs, various deintensification strategies have been tested in clinical trials to maximize response while minimizing toxicity. These approaches have included (1) replacement [10,11], reduction [12,13,14], or omission of cytotoxic chemotherapy [15], (2) de-escalated adjuvant (chemo)radiotherapy following ablative surgery [16,17], (3) reduction in dose of radiation following induction chemotherapy [18,19,20,21,22], and (4) reduction in dose of radiation during definitive (chemo)radiotherapy [23]. Unfortunately, recent phase III clinical trials aimed at de-escalation approaches in HPV+ tumors (De-ESCALate HPV [10], RTOG 1016 [11]) resulted in inferior outcomes and have demonstrated the need for caution in deviation from standard of care [24]. In the wake of these studies, there is a renewed interest in predictive and prognostic biomarkers that may help guide patient selection for treatment tailoring [24,25]. For example, ongoing research on the utility of serum HPV circulating tumor DNA assays [26,27,28,29] has led to promising initial results. Other studies have correlated genomic biomarkers representative of tumor biology subtypes with patient outcomes. These have included p53 status [30], PIK3CA mutation [31], MATH score/ER-alpha expression [32], and TRAF/CYLD loss [33]. Thus, there is a clear need to better understand the biology of HPV-driven carcinogenesis and the profiles and/or reprogramming of key cell subpopulations specific to HPV+ disease.

Given that HPV exclusively infects the mucosal epithelium, and that squamous epithelial cells are the cells of origin for HNSCCs, defining the effects of infection on epithelial subpopulations is essential to advance new insights into HPV-reprogrammed host biology, candidate biomarkers, and targeted therapeutic strategies. Recent advances in single-cell technologies such as single-cell RNA sequencing (scRNAseq) have enabled the bioinformatic identification of distinct subpopulations within heterogenous tissues. With scRNAseq, the transcriptomic signature of distinct epithelial subpopulations can be identified and analyzed to identify their respective gene and pathway signatures. A previous publication [34] described scRNAseq data on HPV+ and HPV− HNSCCs to define distinct cell types, and explored immune-cell-related processes and interactions that shape the tumor immune microenvironment. Herein, we systematically analyzed the corresponding epithelial compartment to identify cellular subpopulations and ontologies enriched in HPV+ vs. HPV− HNSCCs as a foundation for the discovery of novel biomarkers and subtype-specific therapeutic targets.

## 2. Materials and Methods

### 2.1. scRNAseq Data Processing

Publicly available preprocessed single-cell RNA sequencing (scRNAseq) data were downloaded from the Gene Expression Omnibus database: accession ID GSE164690 [34] (Appendix A). The R v4.1.1 library Seurat v4.4 was used for cell-type clustering and marker gene identification [35]. Each sample was normalized by SCTransform, using the glmGamPoi method and the number of RNA molecules per cell was regressed out. Samples were integrated with common anchor genes using the rPCA method to minimize sample-to-sample variation. Cell clusters were determined by the Louvain algorithm. UMAP dimension reduction was performed using the first 30 principal components. Marker genes for each cell type were calculated using the Wilcoxon rank-sum test returning only genes that are present in a minimum of 25% of the analyzed cluster. Epithelial cells were subset by isolating the CD45- cells, reintegrating the data and curating the epithelial cells of interest by using gene expression markers. Original annotations of HPV status, defined based on clinical p16 IHC testing and confirmed by detection of viral genes, were retained for all samples. Global expression of all epithelial cells originating from HPV− versus HPV+ tumors was compared to define globally differentially expressed genes (DEGs). Each scRNAseq cluster was considered a distinct epithelial subpopulation.

### 2.2. Ontology Analysis and Cytoscape Visualization

The transcriptomes and DEGs of select clusters were analyzed using the gProfiler web server [36] for Gene Ontology (GO) terms, which includes Gene Ontology (GO) terms, curated by the Ensembl database [37], and pathways from KEGG (https://www.genome.jp/kegg/, accessed on 1 January 2025), WikiPathways (https://www.wikipathways.org/index.php/WikiPathways, accessed on 3 January 2025), and Reactome (https://reactome.org/, accessed on 4 January 2025). Complex enriched gene sets were visualized using the Enrichment Map App [38] within Cytoscape 3 using parameters previously defined [39]. In brief, enrichment files and gene sets outputs from g:Profiler were processed using Enrichment Map and terms with a false discovery rate less than 0.01 were considered significantly enriched. Nodes represent pathways, and they are connected by edges (lines) between nodes with genes in common. Node size is dependent on the enrichment score associated with that pathway, while edges thickness is relative to the number of genes shared between nodes.

## 3. Results

### 3.1. Identification of Epithelial Cells by Joint scRNAseq Analysis of HPV+ and HPV− HNSCCs

We sought to define the epithelial subpopulations present in HNSCC tumors and their gene signatures, which have not been previously reported. To this end, published scRNAseq data for six HPV+ and nine HPV− HNSCC tumors were processed with batch correction (Figure 1A). UMAPs of all cells split by the HPV status of the parent tumor confirmed a high level of overlap (Figure 1B). Unbiased clustering revealed 20 distinct populations whose ontologies (Figure 1C) were assigned based on the corresponding distinct transcriptomic signatures visualized by heatmap (Figure 1D). The identified cellular populations included immune, epithelial, and stromal cells, demonstrating the heterogeneous cell types present in the tumors.

### 3.2. Identification of Global Transcriptomes and Pathways Distinguished by HPV Status

Keratin-expressing clusters were jointly re-clustered to isolate epithelial cells from other cell types (UMAP Figure 2A). The global signature of epithelial cells in HPV+ vs. HPV− HNSCCs was first analyzed (heatmap, Appendix A). The cytoscape visualization of GO biological processes upregulated in HPV− vs. HPV+ HNSCCs revealed pathways related to immune activation, peptidase activity, and processes such as adhesion, angiogenesis, and apoptosis (Figure 2B). Conversely, a similar analysis for processes upregulated in HPV+ vs. HPV− HNSCCs included pathways related to metabolism, migration, and differentiation (Figure 2C). Additionally, KEGG pathway (Figure 2D) and reactome pathway (Figure 2E) enrichment was assessed. Relative to HPV− HNSCCs, there was an upregulation in HPV+ HNSCCs in processes related to metabolism, tissue development, migration, and signal transduction. In HPV− HNSCCs, there was an upregulation of immune response, cell motility, and peptidase activity, which notably indicated a relative downregulation in immune response activation, cytokine signaling, and T-cell proliferation.

### 3.3. Distinct Epithelial Cell Subpopulations Are Present in HPV− vs. HPV+ HNSCC Tumors

Epithelial subpopulations were identified by unbiased clustering (UMAP Figure 3A, heatmap Figure 3B, dot plot Figure 3C. Cells from both HPV+ and HPV− HNSCCs were represented in all subpopulations. Enrichment (defined as ≥30% difference by cell number) of subpopulations in HPV+ versus HPV− HNSCCs was determined by analyzing the distribution of tumor cells across clusters (Figure 3D, arrows indicating enrichment). C1, C4, C5, C7 were found to be enriched in HPV+ HNSCCs, C2 and C3 in HPV− HNSCCs, and C0 and C6 were roughly equally represented in HPV+ and HPV− HNSCCs. C8 had too few cells to be confidently assessed. Each cluster/subpopulation was defined by a distinct transcriptomic profile, although in some subpopulations, there were differences in the signature between HPV+ and HPV− specimens (heatmap, Appendix A).

### 3.4. HPV+ and HPV− Epithelial Subpopulations Are Defined by Differentiation Status and Cell-Cycle Phase

We determined the differentiation status and cell-cycle phase of epithelial subpopulations as previously described [39]. To this end, we probed for markers of embryonic/basal (COL17A+/KRT5+) (Figure 4A) versus differentiated (KRT13+/DMKN+) (Figure 4B) character, and markers of S (PCNA) (Figure 4C) and G2/M (MKI67) (Figure 4D) phase based on a well-defined set of markers for each [40,41]. This allowed the predominant differentiation state and cell-cycle phase(s) of each epithelial subpopulation to be defined and provide context into major biological processes that may appear in pathway analyses. We find that cells in S phase were predominately in C5, cells in G2/M phase in C3, and cells in C6 were in a mix of S and G2/M phases (Figure 4E). When broken down by HPV status, an increase in cells in S phase was observed for HPV+ vs. HPV− HNSCCs (Figure 4F).

### 3.5. Genes Ontologies of Epithelial Subpopulations Enriched in HPV+ HNSCCs

The C1, C4, C5, and C7 subpopulations were proportionately increased in HPV+ HNSCCs. We have previously discussed C7, which most closely matches the previously reported HIDDEN cell signature, as being upregulated in HPV+ HNSCCs [39]. For each of these subpopulations, we sought to determine the defining processes and gene ontologies to elucidate the identity of these epithelial cell subtypes. We analyzed each of these subpopulations using Cytoscape visualization (Figure 5A, Figure 6A, Figure 7A and Figure 8A) and defined the top hits for GO biological processes (Figure 5B, Figure 6B, Figure 7B and Figure 8B), KEGG pathways (Figure 5C, Figure 6C, Figure 7C and Figure 8D), and reactome pathways (Figure 5D, Figure 6D, Figure 7D and Figure 8C). To identify candidate drivers of each subpopulation, transcription factor binding site enrichment analysis was performed through Transfac analysis (Figure 5E, Figure 6E, Figure 7E and Figure 8E). Altogether, we found that C1 was predominated by mitochondrial-related processes, e.g., cellular respiration and oxidative phosphorylation, C4 by cell migration, p53 signaling, and cancer cell metabolism, C5 by processes related to the S phase of the cell cycle such as DNA replication, and C7 by migration and tissue differentiation.

### 3.6. Ontologies of Epithelial Subpopulations Enriched in HPV− HNSCCs

Similarly, we examined the subpopulations proportionately increased in HPV− HNSCCs, namely, C2 and C3, using Cytoscape (Figure 9A and Figure 10A) and by defining top hits for GO biological processes (Figure 9B and Figure 10B), KEGG pathways (Figure 9C and Figure 10C), and reactome pathways (Figure 9D and Figure 10D). We also determined transcription factor binding site enrichment by Transfac analysis (Figure 9E and Figure 10E). C2 was dominated by mitochondrial-related pathways such as biosynthesis and cellular respiration, signatures shared with HPV+ enriched C1 cells. Lastly, C3 was defined by genes involved in the G2/M cell-cycle regulation, including processes of organelle and chromosome organization, nuclear division, cytokinesis, and nucleic acid metabolism.

### 3.7. Ontologies of Epithelial Subpopulations Equally Represented in HPV− and HPV+ HNSCCs

Two epithelial subpopulations (C0 and C6) were found to be present in HNSCC tumors regardless of HPV status. The same analysis for these subpopulations was performed to define ontologies: Cytoscape (Figure 11A and Figure 12A), GO biological processes (Figure 11B and Figure 12B), KEGG pathways (Figure 11C and Figure 12C), reactome pathways (Figure 11D and Figure 12D), and transcription factor binding site enrichment by Transfac (Figure 11E and Figure 12E). C0 processes predominately related to immune responses, biosynthesis, and cell differentiation, while C6 featured a range of pathways related to the S and G2/M phases of the cell cycle, including ATP synthesis, translation, biosynthesis, and p53 class mediator signaling.

### 3.8. Summary of Newly Identified Transcriptome Signatures

Signatures discovered by Cytoscape visualization, GO biological processes, KEGG pathways, and reactome pathways for each epithelial subpopulation and global expression comparison are summarized in Table 1. Potential avenues for targeting cellular subpopulations based on defining signatures were added.

## 4. Discussion

Ongoing deintensification efforts and the study of new therapies in the treatment of HPV+ and HPV− HNSCCs will benefit from the definition of differences within specific cellular populations and signatures. These offer opportunities for targeted treatment and/or prognostic biomarkers towards improved treatment stratification, especially with the increasing interest in neoadjuvant immunotherapy. Single-cell technologies enable such studies by discovering cellular subpopulations and their respective transcriptomes for greater granularity in our understanding of tumor- and microenvironment-specific HNSCC heterogeneity. Recent studies using scRNAseq have largely focused on differences in immune cell and stromal subpopulations and transcriptomic profiles, which may impact immunotherapy development [34,42]. For example, transcriptome profiles of tumor-infiltrating leukocytes isolated from immunotherapy naïve HPV− vs. HPV+ HNSCC tumors revealed that certain cell types, such as B cells, myeloid cells, and conventional CD4^+^ T cells, have divergent signatures by HPV status [43]. Thus, immunotherapeutic strategies targeting CD8^+^ T cells or CD4^+^ regulatory cells may be equally efficacious between HPV+ and HPV− HNSCCs since these cell types have similar signatures independent of HPV status; however, strategies targeting B cells, myeloid cells, and conventional CD4^+^ T cells could require tailored designs to match the divergent signatures between HPV+ vs. HPV− HNSCCs. These findings highlight the clinical importance of understanding divergent cell subpopulations and signatures of HNSCCs by HPV status.

Because HNSCCs originate in epithelial tissues, we, therefore, reasoned that elucidating the signatures of epithelial subpopulations enriched in HPV+ vs. HPV− tumors may identify biomarkers, as well as drivers of malignancy that might be therapeutic targets. First, the global signature of epithelial cells was defined in HPV+ and HPV− tumors. HPV+ HNSCCs harbored relatively decreased immune response activation signatures, including decreased antigen presentation and interferon signaling. This signature is consistent with numerous reported immunosuppressive activities of the high-risk HPV E5, E6, and E7 oncoproteins [44,45], including the modulation of antigen presentation [46], MHC surface expression [47], and interferon signaling [48]. Additionally, this finding in epithelial cells complements recent scRNAseq data in tumor-infiltrating immune cells, wherein conventional CD4^+^ T cells were found to differ in their differentiation trajectory by HPV status, having downregulated interferon responses and effector memory phenotypes [43]. There was also an upregulation in processes related to epithelial differentiation, consistent with the published disruption of the epithelial differentiation program by HPV towards the formation of an HPV-induced differentiation-dissonant epithelial nonconventional (HIDDEN) surface compartment [39]. Distinct differential signatures related to motility (HPV− HNSCCs) and migration (HPV+ HNSCCs) may reflect the HPV-dependent epithelial-to-mesenchymal transition (EMT) phenotypes. For example, EMT signature genes predicted worse five-year overall survival in HPV−, but not HPV+, HNSCCs [49,50], highlighting underlying mechanisms dependent on HPV. These differences may relate to distinct mechanisms whereby HPV reprograms epithelial cells and drives carcinogenesis, as the E6 and E7 oncogenes have been shown to downregulate E-Cadherin [51] and induce EMT-like processes [52,53]. Lastly, peptidase activity implicated in cancer progression was globally upregulated in epithelial HPV− but not HPV+ cells, indicating therapeutic potential for peptidase inhibitors tailored to HPV− tumors.

The particular advantage of scRNAseq technologies was next leveraged to delineate transcriptomically distinct epithelial subpopulations. Eight subpopulations were subsequently subjected to in-depth analysis. Two of these (C0 and C6) were roughly equally represented; two were enriched in HPV− HNSCCs (C2, C3), and four in HPV+ HNSCCs (C1, C4, C5, C7). First, we examined shared, and, therefore, likely HPV-independent, subpopulations. The C0 subpopulation was defined by metabolic biosynthetic processes while C6 contained cells expressing markers of the G2/M and S phases of the cell cycle, indicative of the HPV-independent presence of cells defined by metabolic and proliferative signatures. Studies of cancer cell metabolism have long highlighted the Warburg effect, unique metabolic rewiring that allows persistent aerobic glycolysis and fuels rapid biosynthesis, growth, and proliferation [54,55]. This signature was indeed observed in the shared C0 subpopulations, and also in the HPV+ enriched C4 subpopulation. In the C6 subpopulation harboring proliferative cells, signaling by p53 class mediators was observed, perhaps representing signaling downstream of the p53 pathway that permits cellular survival and uncontrolled growth. Given that the p53 tumor suppressor is the most commonly mutated gene in HPV− HNSCCs [56] and that high-risk HPV E6 causes the degradation of p53 [57,58,59], this ontology may reflect a shared functional loss of wild-type p53 followed by unrestricted cellular growth, despite the difference in p53 mutational status between HPV− and HPV+ tumors. This ontology was also shared with that of the HPV+ enriched C4 subpopulation. Altogether, standard therapeutic techniques such as chemo/radiation targeting shared proliferation and cancer cell metabolism signatures would be expected to demonstrate efficacy against these HNSCC subpopulations regardless of HPV status.

Next, we examined epithelial subpopulations relatively enriched in HPV− HNSCCs (C2 and C3). The C2 subpopulation was dominated by mitochondrial processes, such as cellular respiration and electron transport chain functions, while C3 consisted of cells in the G2/M phase of the cell cycle, featuring processes involved in organelle and chromosomal organization, nuclear division, and cytokinesis. Future functional analysis of C3 might yield therapeutic approaches targeting mitotic machineries in HNSCC specifically. This is particularly relevant in light of the recent literature elucidating mechanisms and agents to target cancer cell mitochondria as a novel therapeutic [60,61] and/or to improve the response to existing radiotherapy protocols [62]. Interestingly, a potentially analogous subpopulation for C2 was identified in the HPV+ enriched C1 population, which was also defined by mitochondrial and respiration processes. Several mechanisms of mitochondrial reprogramming by HPV have been reported [63], such as the induction of proapoptotic proteins, reactive oxygen species (ROS), and oxidative stress [64]. A comparative analysis of these subpopulations may further define mechanisms whereby HPV reprograms and dysregulates mitochondrial biology and/or ATP production. Differences and similarities between C3 and C1 may then guide similar and/or distinct mitochondrial targeting strategies for both HPV− and HPV+ HNSCCs.

Lastly, the remaining subpopulations enriched in HPV+ HNSCCs were considered. The defining ontologies of C1 (mitochondrial processes), C4 (cancer cell metabolism), and C7 (aberrant differentiation) were previously discussed, leaving the C5 subpopulation consisting of cells in the S phase of the cell cycle. This enrichment reflected a global shift in the distribution of cells defined by cell-cycle markers between HPV+ and HPV− tumors, wherein we observed a notable increase in cells in the S phase (C5) in HPV+ tumors. This finding was expected given that reprogramming of the cell cycle in high-risk HPV-infected SE by the E6 and E7 oncogenes is classically defined by dysplasia, hyperplasia, and hyperproliferation, and infected cells remain dependent on E6/E7 expression upon transformation into SCC. Thus, the enrichment of cells in the S phase of the cell cycle is anticipated for HPV+ tumors and underscores the validity of novel cell populations and their possible functions identified here based on gene expression patterns.

Altogether, this work represents the first step towards a detailed understanding of distinct epithelial subpopulations in HNSCCs, and their defining transcriptomes and associated biological processes as a function of HPV status. In-depth analysis and functional testing of differentially expressed genes, e.g., those regulating mitochondrial processes, may advance the translational goal of developing novel biomarkers and approaches for the tailored or joint suppression of HPV-driven and -independent effectors. Future analysis may permit the discovery of novel biomarkers and therapeutic targets to eliminate epithelial cells that are crucial for sustaining and promoting the development of each tumor subtype, and new treatment approaches that are shared or uniquely dependent on HPV status. A larger sample size of HPV+ tumors will be needed in the future to address the high clinical variability amongst HPV+ HNSCCs and to identify sub-classes of HPV+ tumors that can be used for stratification in de-escalation attempts.

## 5. Conclusions

We identify differentially expressed genes and pathways that define and distinguish HPV+ and HPV– epithelial subpopulations in HNSCC. These can now be used as a foundation for the discovery of biomarkers and, through functional laboratory testing, of effectors of HNSCC progression and therapy resistance. New treatment strategies shared by or tailored to HPV tumor status, both HPV+ and HPV−, may emerge from these data.

## Figures and Tables

**Figure 1 viruses-17-00461-f001:**
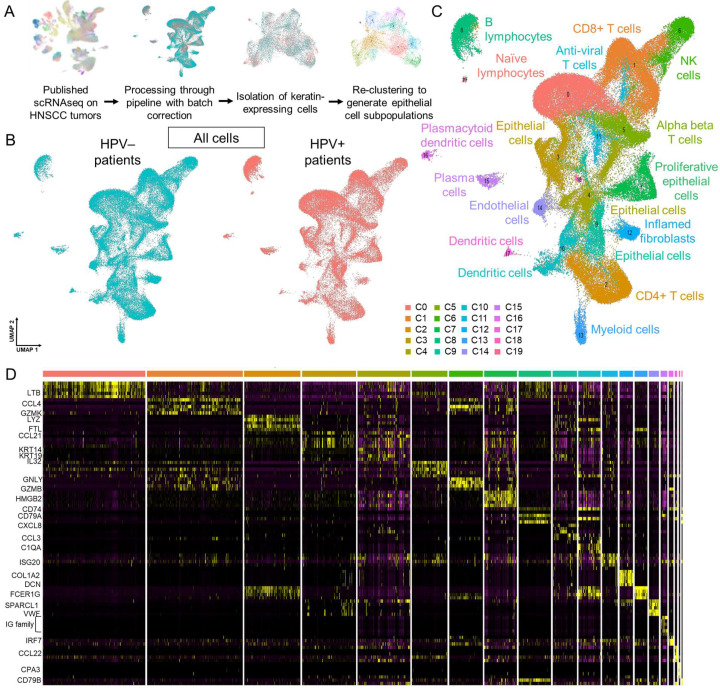
Joint analysis of HPV+ and HPV− HNSCC tumors with batch correction. (**A**) Overview of the bioinformatic pipeline to identify transcriptomically distinct epithelial subpopulations in scRNAseq data of HPV+ (*n* = 6) and HPV− (*n* = 9) HNSCCs. (**B**) Resulting UMAP of all cells separated by patient HPV status. (**C**) Cell clustering demonstrates a mix of immune, epithelial, and stromal cell types. (**D**) Heatmap of transcriptomically distinct clusters with select genes labeled.

**Figure 2 viruses-17-00461-f002:**
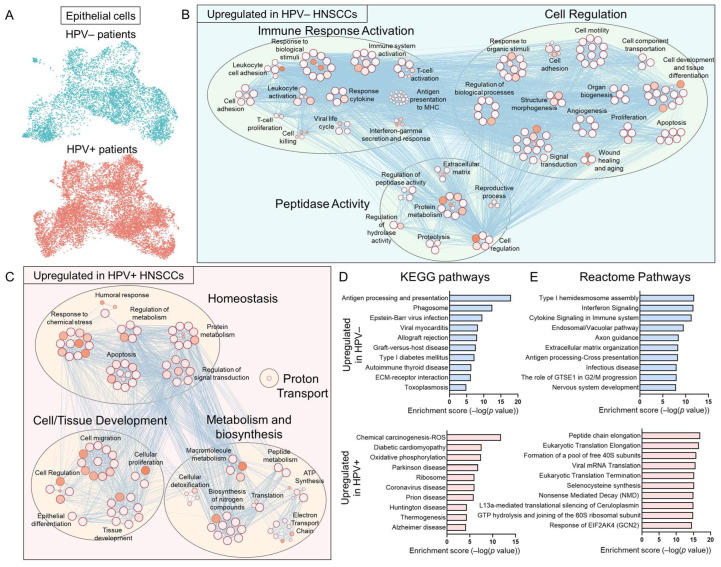
Signatures upregulated in epithelial cells from HPV+ and HPV− HNSCCs. (**A**) UMAP of keratinocytes isolated from the published scRNAseq data and re-clustered, separated by HPV status to show extensive overlap. (**B**–**D**) Global differential expression analysis by HPV status. (**B**) Cytoscape visualization of GO biological processes upregulated in HPV− vs. HPV+ HNSCCs reveals pathways related to immune activation, peptidase activity, and processes such as adhesion, angiogenesis, and apoptosis. (**C**) Cytoscape visualization of GO biological processes upregulated in HPV+ HNSCCs includes pathways related to metabolism, migration, and differentiation. Top 10 KEGG pathways (**D**) and reactome pathways (**E**) upregulated in HPV− and HPV+ HNSCCs.

**Figure 3 viruses-17-00461-f003:**
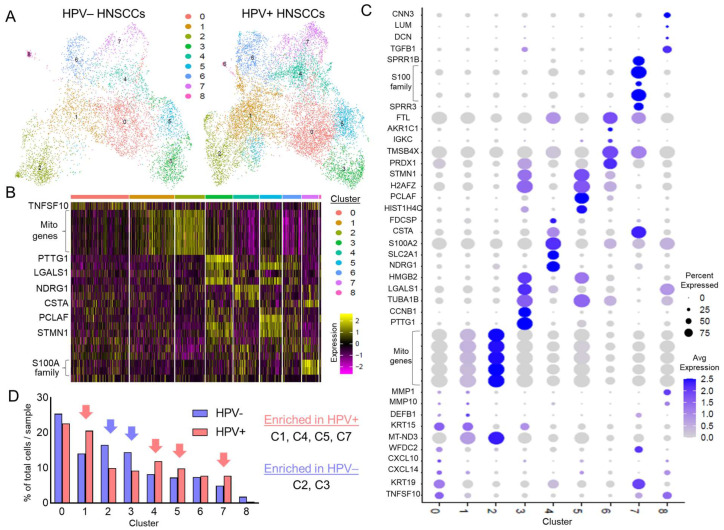
Enrichment of distinct epithelial subpopulations in HPV+ and HPV− HNSCC. (**A**) UMAP showing transcriptomically distinct clusters representing distinct epithelial subpopulations, separated by HPV status of HNSCC tumor. (**B**) Heatmap corresponding to clusters with select genes labeled. (**C**) Dot plot of top genes expressed per cluster. (**D**) Distribution of cells from HPV+ and HPV− HNSCCs across scRNAseq clusters, with those enriched ≥30% by cell counts highlighted by arrows.

**Figure 4 viruses-17-00461-f004:**
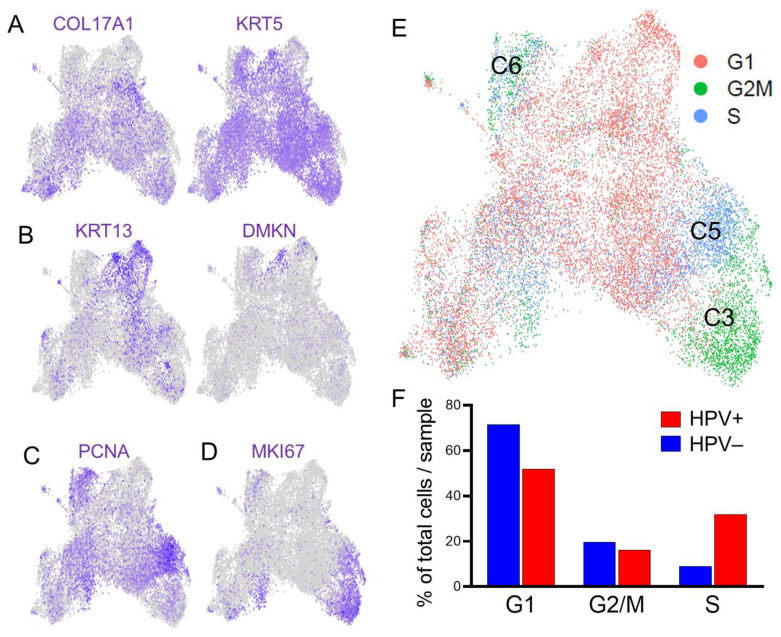
Analysis of differentiation status and cell-cycle phase in epithelial subpopulations. Feature plots demonstrating distribution of select basal (**A**), differentiated (**B**), S phase (**C**), and G2/M phase (**D**) markers. (**E**) Distribution of G1, G2/M, and S-phase cells. C5 harbors predominately cells in S, C3 cells in G2/M, and C6 a mix of S and G2/M cells. (**F**) Distribution of epithelial cells across cell-cycle phases by HPV status.

**Figure 5 viruses-17-00461-f005:**
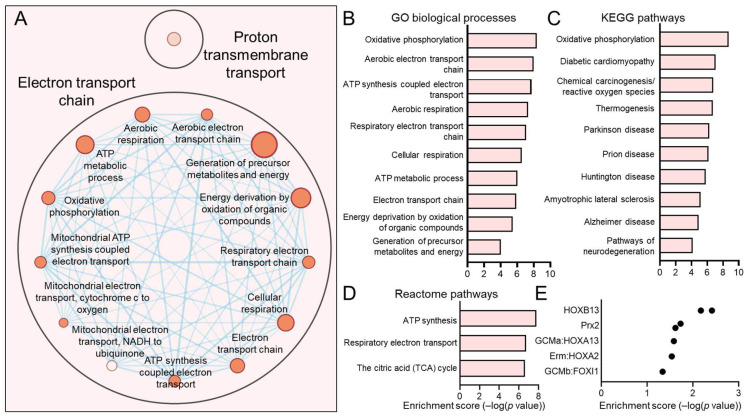
Gene ontologies of the HPV+ enriched C1 HNSCC epithelial subpopulation. (**A**) Cytoscape visualization of GO biological processes highlight genes related to the electron transport chain and aerobic respiration. Summary of top GO biological processes (**B**), KEGG pathways (**C**), and reactome pathways (**D**) demonstrates that the transcriptome of C1 centers on mitochondrial processes and ATP generation. (**E**) Transfac analysis for transcription factor drivers of the C1 subpopulation.

**Figure 6 viruses-17-00461-f006:**
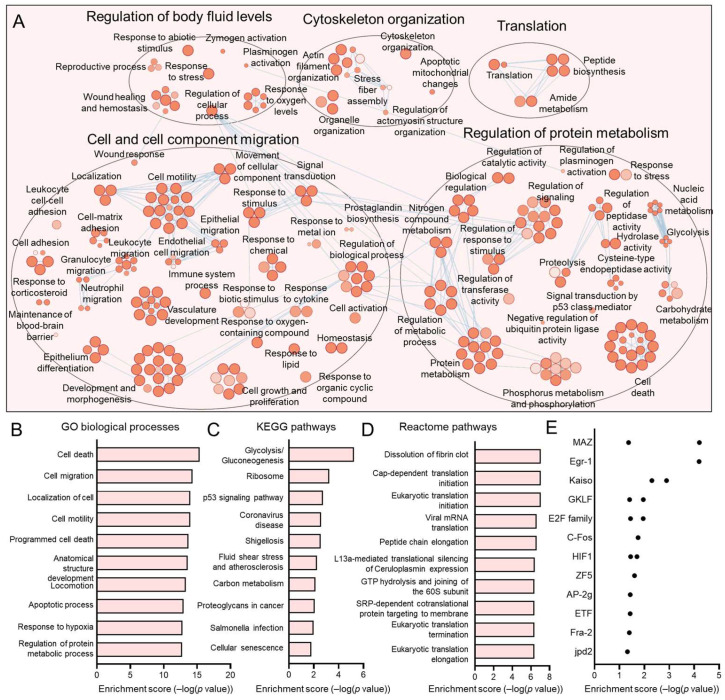
Gene ontologies of the HPV+ enriched C4 HNSCC epithelial subpopulation. (**A**) Cytoscape visualization of GO biological processes highlights pathways related to migration, cytoskeleton organization, and metabolic processes. Summary of top GO biological processes (**B**), KEGG pathways (**C**), and reactome pathways (**D**). (**E**) Transfac analysis for transcription factor drivers of the C4 subpopulation.

**Figure 7 viruses-17-00461-f007:**
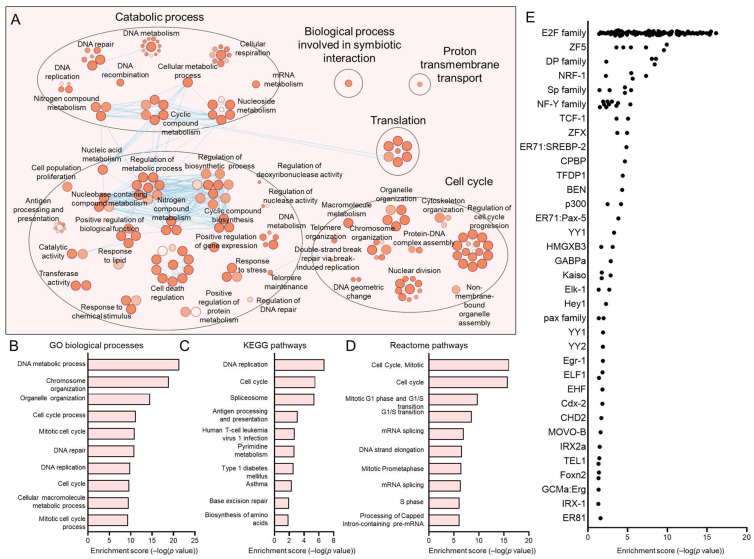
Gene ontologies of the HPV+ enriched C5 HNSCC epithelial subpopulation. (**A**) Cytoscape visualization of GO biological processes highlights pathways related to proliferation. Summary of top GO biological processes (**B**), KEGG pathways (**C**), and reactome pathways (**D**) centers on cell-cycle processes, particularly S phase. (**E**) Transfac analysis for candidate transcription factor drivers of the C5 subpopulation.

**Figure 8 viruses-17-00461-f008:**
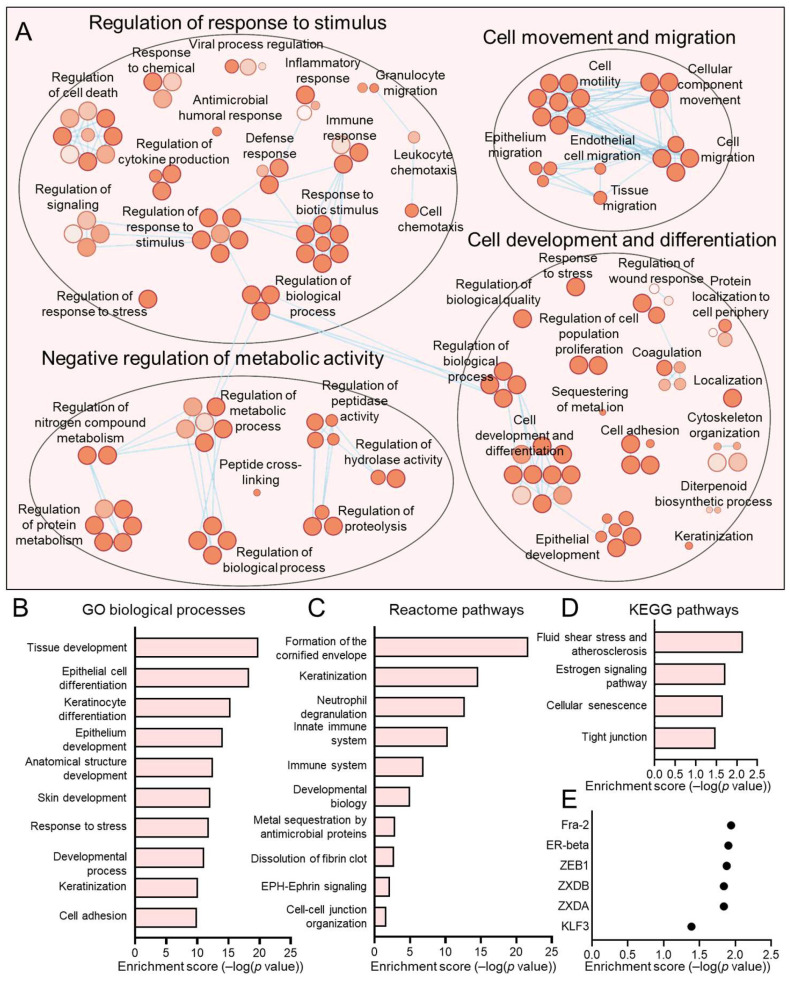
Gene ontologies of the HPV+ enriched C7 HNSCC epithelial subpopulation. (**A**) Cytoscape visualization of GO biological processes highlights pathways related to migration, development, and metabolism. Summary of top GO biological processes (**B**), reactome pathways (**C**), and KEGG pathways (**D**) include keratinocyte differentiation related processes. (**E**) Transfac analysis discover candidate transcription factor drivers of the C7 subpopulation.

**Figure 9 viruses-17-00461-f009:**
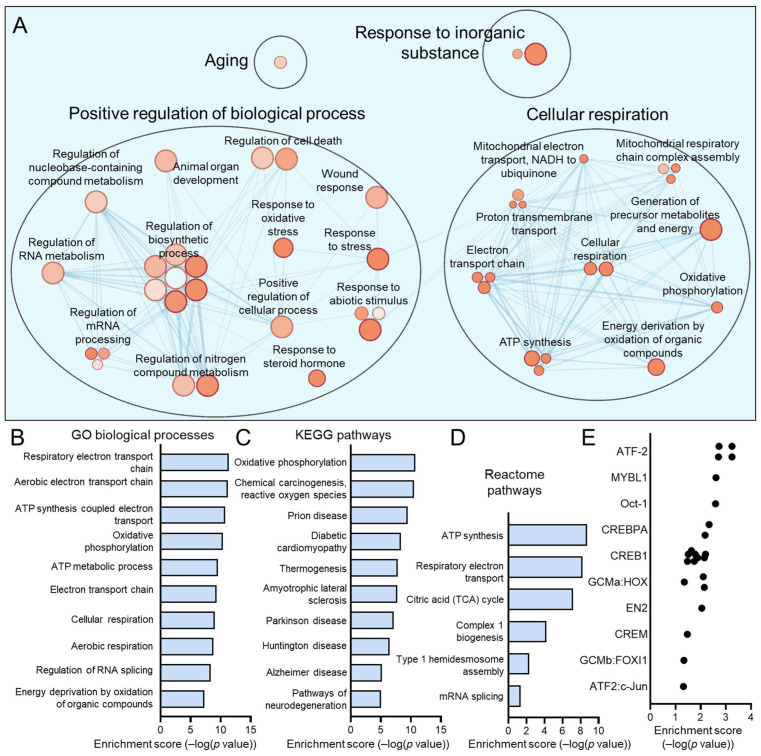
Gene ontologies of the HPV− enriched C2 HNSCC epithelial subpopulation. (**A**) Cytoscape visualization of GO biological processes highlights pathways related to biosynthesis and cellular respiration. Summary of top GO biological processes (**B**), KEGG pathways (**C**), and reactome pathways (**D**) centers on mitochondrial and ATP metabolic processes. (**E**) Transfac analysis for transcription factor drivers of the C2 subpopulation.

**Figure 10 viruses-17-00461-f010:**
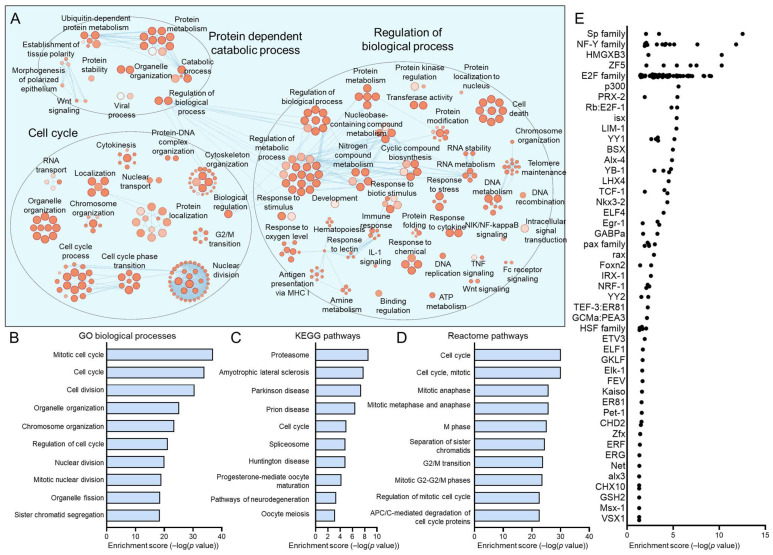
Gene ontologies of the HPV− enriched C3 HNSCC epithelial subpopulation. (**A**) Cytoscape visualization of GO biological processes highlights pathways related to cell cycle and proliferation. Summary of top GO biological processes (**B**), KEGG pathways (**C**), and reactome pathways (**D**) centers on the G2/M phase of the cell cycle. (**E**) Transfac analysis for transcription factor drivers of the C3 subpopulation.

**Figure 11 viruses-17-00461-f011:**
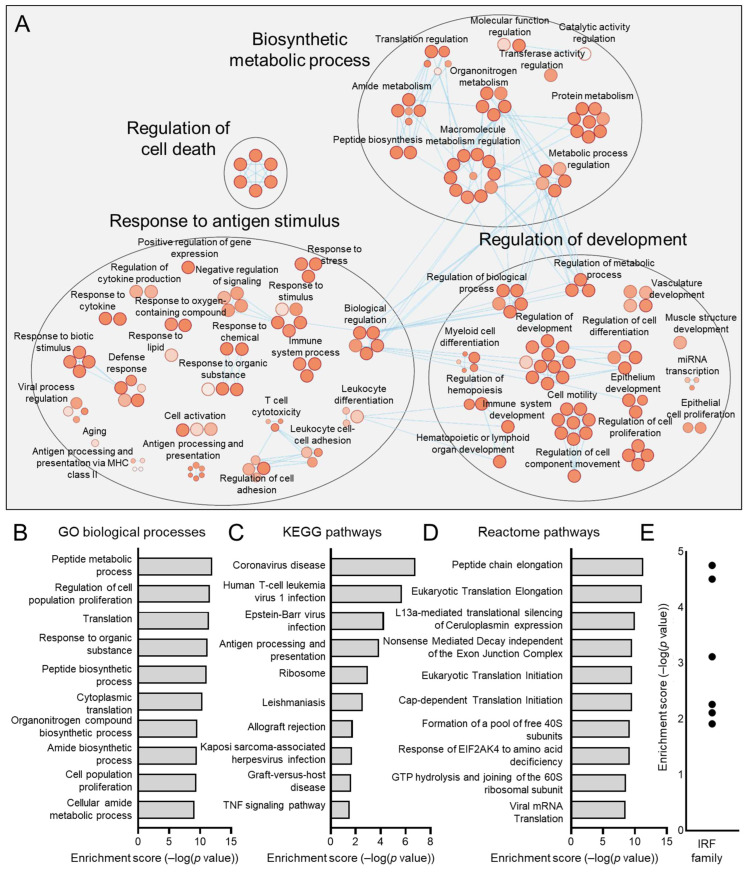
Gene ontologies of the shared C0 HNSCC epithelial subpopulation. (**A**) Cytoscape visualization of GO biological processes highlights pathways related to immune response and proliferation. Summary of top GO biological processes (**B**), KEGG pathways (**C**), and reactome pathways (**D**) centers on antigen processing and translation. (**E**) Transfac analysis for transcription factor drivers of the C0 subpopulation.

**Figure 12 viruses-17-00461-f012:**
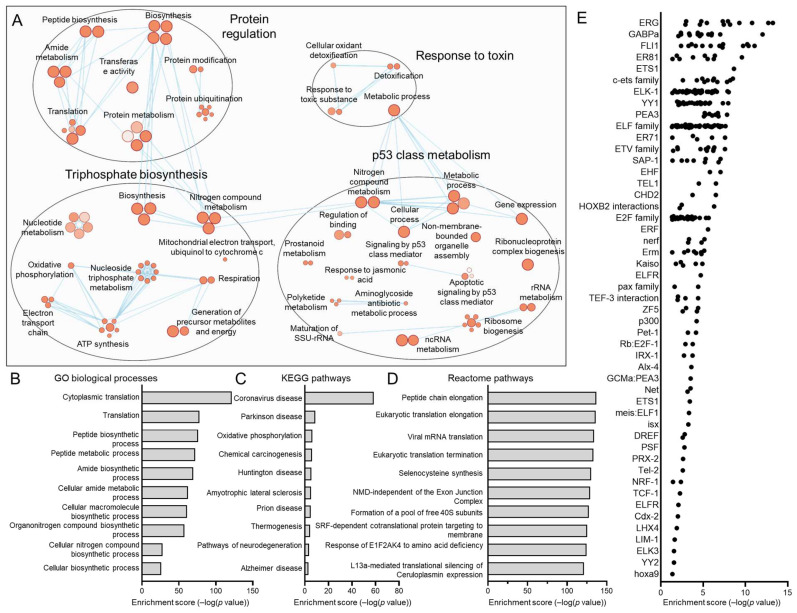
Gene ontologies of the shared C6 HNSCC epithelial subpopulation. (**A**) Cytoscape visualization of GO biological processes highlights pathways related to biosynthesis and p53. Summary of top GO biological processes (**B**), KEGG pathways (**C**), and reactome pathways (**D**) centers on translation and biosynthesis. (**E**) Transfac analysis for transcription factor drivers of the C6 subpopulation.

**Table 1 viruses-17-00461-t001:** Summary of select signatures and potentially targetable phenotypes of cellular subpopulations identified by scRNAseq analysis.

	Enrichment	Select Signatures	Candidate Targetable Phenotypes
Global	HPV−	- Immune response - Cell motility - Peptidase activity	- Peptidases
Global	HPV+	- Biosynthesis - Epithelial differentiation - Migration - Signal transduction	- Metabolic pathways - Immune reactivation - Plasticity
C0	Shared	- Biosynthesis - Antigen processing and presentation - Cell differentiation	- - Metabolic pathways - - Immune reactivation
C6	Shared	- S and G2/M phase of cell cycle - Biosynthesis - Oxidative phosphorylation - Signaling by p53 class mediators	- Cell-cycle checkpoints - Mitochondrial processes - p53 signaling
C2	HPV−	- Cellular respiration - Electron transport chain	- Mitochondrial processes
C3	HPV−	- G2/M phase of cell cycle - Organelle and chromosome organization - Nuclear division and cytokinesis - RNA/DNA metabolism	- Cell division - Mitotic machinery - Nucleotide metabolism
C1	HPV+	- Oxidative phosphorylation - Electron transport chain	- Mitochondrial processes
C4	HPV+	- Cell migration - Protein metabolism - p53 signaling - Glycolysis/gluconeogenesis	- Invasion - p53 signaling - Cancer cell metabolism
C5	HPV+	- S phase of cell cycle - DNA replication - Chromosome and cytoskeletal organization - Translation	- DNA replication machinery - Translation machinery
C7	HPV+	- Cell migration - Epithelial differentiation	- Aberrant HPV-driven differentiation (HIDDEN cells)

## Data Availability

Data are contained within the article and Appendix A.

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
