# Peer review of "A Single-Cell Transcriptome Atlas of Epithelial Subpopulations in HPV-Positive and HPV-Negative Head and Neck Cancers"

_viruses, 2025, doi:10.3390/v17040461_

Round 1
Reviewer 1 Report
Comments and Suggestions for Authors
1. In the introduction you write "It is well established that HPV+ and HPV- HNSCCs differ in anatomical distribution." However, how can HPV influence anatomical distribution? This needs to be either removed or spelled out and explained what the authors meant.
2. You write "education in a dose of radiation during chemotherapy." However, the question arises, what could be the dose of radiation during chemotherapy? Perhaps you meant radiotherapy? Or the dose of chemotherapeutic drugs? (lines 60-61).
3. 58% of the literature sources are old publications (they are more than 5 years old). Old articles need to be replaced with more recent ones.
Dear author, I do not consider myself qualified enough and able to evaluate the quality of the English language in this work. However, based on my own experience, I can say that I have no comments on the quality of the English language.
Reviewer 2 Report
Comments and Suggestions for Authors
Article should be improved

See comments
Reviewer 3 Report
Comments and Suggestions for Authors
Human papillomavirus (HPV) has long been recognized as being responsible for a high percentage of head and neck squamous cell carcinomas (HNSCC). Indeed, the incidence of HPV+ HNSCCs is increasing due to the high incidence of oral HPV infection. However, there is also a significant percentage of these cancers that are HPV- and HPV+ and HPV- cancers differ in multiple ways. Indeed, these differences call for the use of distinct treatment strategies in order to achieve the best outcome in a timely manner.
In this manuscript, the authors analyze published single cell RNA sequencing (scRNAseq) data of HPV+ and HPV- HNSCCs to try to identify genes that are differentially expressed in the two epithelial subpopulations. The goal is to use these data to inform the identification of biomarkers and pathways distinct to either HPV+ or HPV- cancers that can then serve as targets for subsequent therapeutic approaches that specifically target them.
The authors present a persuasive, compelling case for their achievement of this goal. Analysis of the scRNAseq data identifies transcriptomes and pathways specific to either HPV+ or HPV- HNSCCs. For example, the following signatures are only a few of those identified as being enriched in HPV- cells: immune response, cell motility, cellular respiration, electron transport chain, etc. For HPV+ cells, these signatures include biosynthesis, epithelial differentiation, migration, oxidative phosphorylation, etc. The authors go on to identify candidate targets for each specific signature, which virtually lays out the roadmap for a plethora of experimental approaches to identify different susceptible target pathways or markers specific for either HPV+ or HPV- HNSCCs. This is considered a major strength of the study, as it definitively establishes the potential of these findings.
This is considered a beautifully constructed, highly illuminating presentation of an extremely large data set. The authors do a tremendous job of, not only effectively displaying the data, but also subjectively interpreting it. As such, the study is considered a major contribution to the HNSCC field, one that identifies an abundance of novel avenues of investigation with the potential to overcome the differences in HPV+ and HPV- cancers that confound the development of a single treatment that is efficacious against both. One can easily envision these findings eventually leading to the development of effective, but different, approaches to control these diseases.
